# Buprenorphine Versus Methadone in Female New Zealand White Rabbits Undergoing Balanced Anaesthesia for Calvaria Surgery

**DOI:** 10.3390/ani15131843

**Published:** 2025-06-22

**Authors:** Daniela Casoni, Chiara Parodi, Luisana Gisela Garcia Casalta, Kay Nettelbeck, Claudia Spadavecchia

**Affiliations:** 1Experimental Surgery Facility, Experimental Animal Center, University of Bern, 3010 Bern, Switzerland; chiara.parodi@unibe.ch (C.P.); luisana.garcia@unibe.ch (L.G.G.C.); kay.nettelbeck@unibe.ch (K.N.); 2Atrial Fibrillation & Therapeutic Innovation, Department of Physiology, University of Bern, 3012 Bern, Switzerland; 3Anaesthesia Division, Department of Clinical Veterinary Science, Vetsuisse Faculty, University of Bern, 3012 Bern, Switzerland; claudia.spadavecchia@unibe.ch

**Keywords:** analgesia, rabbits, bone surgery, opioids

## Abstract

The administration of analgesia in rabbits is very much linked to anecdotal beliefs and personal experience. Experimental evidence is very limited; this poses a real problem for anaesthesia specialists who face the growing demand of suggesting efficacious solutions to counteract pain in rabbits undergoing invasive surgeries. The challenge is particularly perceived in research animals, which serve the fundamental purpose of increasing medical knowledge to the benefit of several species but the use of which imposes the moral obligation of administering the best analgesia care. This study aimed at adding a piece in this complex puzzle: we compared the pros and cons of two analgesics (buprenorphine and methadone) combined with other drugs to allow a pain-free surgical experience in female rabbits undergoing bone surgery on the head. Both drugs can be used in healthy rabbits without clinical side effects and with good efficacy at the doses injected. We thought that methadone would have brought about a better analgesia than buprenorphine, but we observed that buprenorphine guaranteed a profounder and longer pain-free experience. Some animals needed more analgesics than others, underlining the fact that pain is an individual experience, and rescue drugs must be titrated based on repeated pain assessment and scoring.

## 1. Introduction

The rabbit is an increasingly popular household pet, and a common animal model in biomedical research. In 2022, more than 370,000 rabbits were used in Europe and Norway for experimental purposes [1]. In biomedical research, among other uses, rabbits are frequently chosen for development of new surgical techniques, which might be invasive and associated with an unpredictable degree of pain. Rabbits are suitable for the screening of bone implants and development of new materials for bone replacement [2]. This implies that they often undergo orthopaedic procedures which, according to veterinarians’ opinions, are the most painful procedures in rabbits [3]. Pain must be prevented or treated in experimental animals for ethical and legal reasons (Animal Welfare Act of 2005 and Animal Protection Ordinance). Opioids are considered the mainstay for the management of moderate to severe pain; however, their usage is not free from side effects. Sedation, respiratory depression, and reduced gastrointestinal motility are often reported in mammals. In light of its long duration of action and its partial μ-agonist mechanism of action, buprenorphine is one of most frequently used opioids in rabbits, and the far most used in laboratory rabbits [4]. It is commonly assumed that its partial μ-agonism brings about a lower incidence of unwanted effects compared to pure µ-agonists. However, experimental evidence to support this is limited [5]. Furthermore, the preoperative use of buprenorphine is questionable. Indeed, its bell-shaped dose–response curve makes buprenorphine potentially insufficient to counteract moderate to severe pain, while its high affinity to the μ-receptors might interfere with the subsequent administration of a pure μ-agonist if rescue intraoperative analgesia would be needed [6]. A previous report in male New Zealand white rabbits highlighted the invasiveness of the calvaria surgery and the limitation of preoperative buprenorphine in guaranteeing postoperative analgesia [7]. Although rabbits should be monitored for undesirable effects of opioids, pain itself can also result in respiratory depression, abnormal behaviour, depressed food intake, and ileus [5]. Therefore, concerns for the occurrence of side effects should not result in the withholding of pure μ-agonist opioids. Methadone, a representative of the diphenylheptylamines, is an opioid with high and selective affinity for µ-receptors [8] and the ability to interact with the N-methyl-D-aspartate receptor, which contributes to its analgesic effect [9]. Very little is known about methadone’s use with rabbits. Previous studies reported that a subcutaneous injection of 2 mg Kg^−1^ provided sedation and increased pressure pain thresholds in conscious rabbits, without interfering with the faecal production [10] and that doses of 0.2–0.3 mg kg^−1^ SC or IM did not impair gastro-intestinal motility and appetite [11,12]. It is a shared and accepted concept among anaesthesiologists that providing analgesia before the application of a painful stimulus may reduce neural responsiveness to noxious inputs, as well as postoperative pain and, thus, analgesic drug requirements [5].

In the context of a balanced anaesthesia protocol, it is currently unknown whether buprenorphine or methadone might have different respiratory effects and sedative and analgesic profiles when administered in premedication to rabbits undergoing invasive surgeries.

The primary aim of this study was to assess and compare the intra- and postoperative analgesic efficacy of buprenorphine and methadone in rabbits undergoing calvaria surgery. Secondary aims were to assess and compare the degree of sedation and respiratory depression elicited by the two opioids. We hypothesised that methadone would elicit deeper sedation and more effective intraoperative analgesia than buprenorphine, with no difference in respiratory depression.

## 2. Materials and Methods

### 2.1. Ethical Approval

The study was revised and approved by the Committee for Animal Experiments of the Canton of Bern, Switzerland (national number 29384). In Switzerland, animal experimentation is strictly regulated by the Animal Welfare Act of 2005 and the Animal Protection Ordinance.

### 2.2. Animals

Forty-nine healthy (ASA 1) female New Zealand white rabbits (NZWRs) undergoing calvaria surgery between September 2018 and October 2019, with a minimum weight of 3 kg and a minimum age of 3 months, were enrolled in the study. The study was carried out at the University of Bern. Female rabbits were selected for the calvaria surgery as a previous study in males was carried out.

The rabbits were housed in groups (3–4 compatible subjects) in the Central Animal Facility (CAF–Experimental Animal Center, University of Bern) on elevated cages (3 R-type cage, Techno Plast, floor area of 4670 cm^2^ put in communication, hosted a group). At least two weeks of acclimatisation after importation from France (Charles River Laboratories, Domaine d’Oncin, Lyon, France) were guaranteed before the surgical intervention. The facility guaranteed an adjusted climate (temperature of 22–24 °C ± 2 °C, humidity of 30–60% ± 5%) and special sun substitution with UV light (photoperiod of 6–18 h). Environmental enrichment was provided with hay and toys. Water and pellet food was accessible ad libitum and not withheld before anaesthesia. During the acclimatisation, rabbits were used to frequent human handling, including petting and putting into transport boxes.

The morning of the surgery, rabbits were transferred using an on-purpose transport box, where they could maintain visual and auditory contact with a co-mate, from the CAF to the experimental surgery facility (ESF), where anaesthesia and surgery were carried out. At the end of the surgery, the rabbits were placed in individual recovery cages under constant veterinary supervision. The rabbits were transported back to the Central Animal Facility when deemed pain-free and able to drink and eat independently.

### 2.3. Anaesthesia Management

Rabbits were equally randomly assigned either to group M (methadone) or to group B (buprenorphine) with a randomisation generator [13]. The veterinary anaesthesiologist responsible for anaesthesia and data collection was unaware of the treatment. After the transport, the rabbits were allowed to interact freely on the floor with the anaesthetist, who approached them gradually and accustomed them to the auscultation, head touching, and petting. Rabbits were held according to the recommendations of Bradbury and Dickens [14]. Afterwards, clinical examination, including weighing, was carried out, and, in preparation for arterial and venous cannulation, EMLA cream was topically applied on the external ear auriculae. Thereafter, 0.1 mg kg^−1^ of dexmedetomidine (Dexdomitor^®^, 0.5 mg mL^−1^, VETOQUINOL Gmbh, Ismaning, Germany), 15 mg kg^−1^ of ketamine (Narketan^®^, 100 mg mL^−1^, VETOQUINOL Gmbh, Germany), and either 0.03 mg kg^−1^ (group B) of buprenorphine (Temgesic^®^, 0.3 mg mL^−1^, Schering-Plough, Lucerne, Switzerland) or 0.3 mg kg^−1^ (group M) of methadone (Methadon Inj Lös, 10 mg mL^−1^, Streuli Pharma AG, Uznach, Switzerland) mixed in the same syringe were injected subcutaneously in the dorsal thoracic region. Animals were left undisturbed in their box for 10 min, and afterwards, they were supplemented with oxygen through a non-tight face mask (3 L min^−1^). Fifteen minutes after the injection, the degree of sedation was assessed through a score sheet adapted from Raekallio et al. [15] (see Appendix A). If sedation was scored >7/11, the rabbits were lifted from the box and put on the preparation table; otherwise, they were left undisturbed for further 5 min and reassessed. If the sedation score remained ≤7/11, a supplementary dose of ketamine (5 mg kg^−1^) and dexmedetomidine (0.02 mg kg^−1^) was injected intramuscularly in the triceps muscle. After 5 min, scoring was repeated, and the animals were lifted onto the table. Once on the table, whilst oxygen supplementation was continuously provided, a 22 or 24 G cannula was introduced into one of the marginal auricular veins, and a 22 G cannula was introduced into one auricular artery. The eyes were lubricated with Vitamin A ointment. Antibiotic prophylaxis consisted of a combination of procaine and benzathine penicillin at dosage of 30,000 I.U. kg^−1^ SC (Duplocillin^®^LA, MSD Animal Health GmbH, Lucerne, Switzerland). Ropivacaine (0.75%) diluted 1:2 was administered beside the incision line intradermally after clipping and disinfecting the surgical field and as a single deposit into the periosteum for a maximum of 3 mg kg^−1^. Auto-adhesive ECG electrodes were put on the paws of right and left forelimbs and left rear limb. After transferring the animals to the operating theatre, a laryngeal mask was introduced under the guidance of the capnography curve. If the depth of anaesthesia was insufficient to allow it, a bolus of midazolam (Dormicum^®^, 5 mg 5 mL^−1^, Cephalpharm Schweiz Gmbh, Binningen, Switzerland) (0.2 mg kg^−1^) was injected IV. Ringer lactate infusion (5 mL kg^−1^ h^−1^) was started thereafter and continued until the end of the surgical procedure. General anaesthesia was maintained with isoflurane in oxygen targeting an end-tidal isoflurane of 1.3% administered through a modified Jackson Rees T-system with 0.5 L closed tail bag. The rabbits were allowed to breathe spontaneously, but manual ventilation was provided temporarily for resolving apnoeas longer than 30 s. During general anaesthesia, continuous monitoring of heart rate, respiratory rate, oxygen arterial saturation, capnography, invasive blood pressure, oesophageal temperature, and inspired and expired fraction of gasses (air, CO_2_, and isoflurane) was provided via a multi-parametric monitor (Carescape B850, Anandic Medical Systems AG, Feuerthalen, Switzerland). Noninvasive blood pressure was measured with doppler technique applying the probe (Eickemeyer^®^ Ultrasonic Doppler, Eickemeyer Medizintechnik für Tierärzte AG, Appenzell, Switzerland) above the plantar digital artery. An arterial blood sample was collected and analysed after the first surgical incision.

Intraoperative nociception was assessed continuously through cardiorespiratory monitoring and toe pinch reaction tested every 5 min. Nociception was deemed insufficient in presence of autonomic responses to surgical stimuli and/or presence of a positive toe pinch reaction. In particular, if an increase of ≥20% was observed in at least 1 out of 3 variables (mean arterial blood pressure, respiratory rate, and heart rate) compared to pre-incisional values, and/or toe pinch was positive, ketamine (2 mg kg^−1^) was administered IV as rescue intraoperative analgesia. To prevent or counteract hypothermia, an air force warming device was used. If hypotension (defined as MAP < 60 mmHg) occurred, noradrenaline was titrated (0.01–0.03 µg kg^−1^ min^−1^) to achieve an MAP of 60–70 mmHg. At the end of the surgical procedure, isoflurane was discontinued, and the supraglottic device was removed when palpebral reflexes and swallowing reflex were resumed. Meloxicam (0.5 mg kg^−1^) was injected IV into all rabbits.Atipamezole (Antisedan^®^, 5 mg mL^−1^, VETOQUINOL Gmbh, Germany)was injected IM at a dosage of 0.4 mg Kg^−1^ if the time between the removal of laryngeal mask and the time to achieve sternal recumbency exceeded 60 min.

### 2.4. Surgery

A 3.5 cm incision was made from the nasal bone to the mid-sagittal crest. The parietal bone was exposed following the elevation of the periosteum, and two 10 mm diameter bone defects for each parietal bone were prepared with a trephine and diamond round burs under copious irrigation with sterile saline. Maximal care was taken to avoid injury of the dura mater. Different materials were then implanted into the defects (a total of 600 µL of arterial blood was used to mix with each granule of the tested materials) while 300 µL of unmixed arterial blood was filled up into the negative control (sham). After implantation of the materials, a 12.5 mm × 13.0 mm resorbable collagen membrane was used to cover the defect site. The periosteum and skin were closed with interrupted sutures in layers using 4–0 Vicryl^®^ and 4–0 Monocryl^®^ sutures (Ethicon, Somerville, NJ, USA). The wound surface was sealed with a spray film dressing (OPSITE^®^ SPRAY, Smith & Nephew, London, UK).

### 2.5. Postoperative Management and Pain Assessment

After removal of the laryngeal mask, the rabbits were hosted in a cage, and temperature was controlled with a forced-air warming device. Supplemental oxygen was provided through a non-tight mask at a flow of 3 L min^−1^, and monitoring of SpO_2_ and pulse rate was performed continuously through a pulse oximeter until the rabbits spontaneously achieved sternal recumbency. Time of spontaneous sternal recumbency was defined as the moment at which the animals could maintain the sternal position without external support, and they could hold their head. Thirty minutes after the removal of the laryngeal mask, an arterial blood sample was retrieved from the arterial catheter, and blood gas analysis was performed.

From this time on, monitoring consisted of hourly to bihourly recording of heart rate, respiratory rate, and rectal temperature until the animals recovered fully and were able to hop in the cage.

Pain assessment was performed by evaluation through the Rabbit Grimace Scale following direct visual observation of the animals and testing peri-incisional allodynia using Von Frey filaments. Using the same evaluator, an experienced anaesthesiologist blinded to the treatment groups (DC), performed all the pain evaluations. A baseline pain assessment was carried out before any drugs were administered on the morning of the surgery, once the rabbits could accept head touching. Postoperative pain assessments were then carried out at intervals of 2 h starting from the achievement of sternal recumbency, for a total of at least 5 postoperative evaluations on the surgical day. The anaesthetist in charge of the evaluations (DC) always remained at a distance from the cage and out of the visual field of the animals. Once RbtGS scores were attributed, the rabbits were gently approached, and mechanical allodynia was tested.

For testing allodynia, Von Frey monofilaments (BioSeb Lab, Vitrolles, France) from 0.008 to 15 g (cm^2^)^−1^ were used. The monofilaments were applied at approximately 1 cm from the incision on three points along the same line on the right and on the left side of the incision and from the thinnest to the thickest. Every evaluation was repeated twice for a total of 12 evaluations per filament (see Figure 1). The Von Frey filaments were positioned on the skin at a 90° angle and pressed until warped; then, they were kept in position for 1.5 s and finally retracted. A positive response was defined as reacting twice to the same filament, or a single reaction to a filament, confirmed at the subsequent monofilament. In the latter case, the highest value was considered as reliable. The threshold was then defined as the pressure in g (cm^2^)^−1^ provoked by the filament at which the response was observed. The test was discontinued when a positive response was elicited or when no reaction was recorded at 15 g (cm^2^)^−1^. In the last case, the test was deemed negative, and the animal was classified as non-responder. Von Frey responders were instead defined as the animals that showed a reaction when tested with the filaments in the range of 0.008 to 15 g (cm^2^)^−1^. The determination of this cut-off was due to the authors’ observation that naïve female rabbits of the same breed and size reacted to the application of filaments thicker than 15 g (cm^2^)^−1^ independently of the presence of pain. A positive reaction was defined as a purposeful head retraction or a body escape movement following buckling of the filament.

Buprenorphine was administered SC at 0.02 mg kg^−1^ if the RbtGS score was ≥4/10 at any assessment. If buprenorphine was not sufficient to obtain a score < 4 within two hours, a second dose of meloxicam was administered IV (0.5 mg kg^−1^).

Once discharged from the experimental surgery facility, the rabbits received meloxicam (0.5 mg kg^−1^ IV) for 3 further days, and further analgesia was provided according to a designed score sheet.

### 2.6. Sample Size Calculation and Data Management

Based on clinical experience with other species, it was initially assumed that 80% of rabbits in group B and 40% of rabbits in group M would have required, at least once, the administration of rescue analgesia during surgery. A minimum of 16 rabbits per group was deemed to be necessary, with a β-error of 0.2 and an α-error of 0.05, to demonstrate a statistically significant difference between groups [16]. An ad interim analysis performed by an independent assessor (KN) revealed that the difference between groups was much lower than expected. Thus, in order to avoid a highly underpowered study, and in absence of safety concerns, all the 49 animals scheduled to undergo surgery for the purpose of the main experiment were included in the present trial.

Statistical analysis was conducted with Sigma Plot 13.0 (Systat Software GmbH, Erkrath, Germany) and NCSS Statistical Software 2024 (NCSS, Silver Spring, MA, USA). Normality of the numerical continuous data was assessed with the Shapiro–Wilk test. Data normally distributed and passing the equal variance test (weight, anaesthesia duration, time elapsing between removal of laryngeal mask, and achievement of sternal recumbency) were compared between group B and group M with *t*-test. Data non-normally distributed in at least one group (blood gas analyses) were compared between group B and group M with the Mann–Whitney rank sum test. Ordinal data (sedation score, RbtGS, Von Frey filaments) were compared between group B and group M with the Mann-Whitney rank sum test. For the outcome RtGS and Von Frey filaments, the test was repeated for each time point. Animals receiving rescue analgesia at one time point were tested as planned at every consecutive time point, but the results of the following time points were excluded from further analyses. A value of 26 g (cm^2^)^−1^, i.e., the value of the next possible Von Frey filament, was attributed to the rabbits that did not respond to the thickest filament tested. To assess modifications of RtGS and Von Frey tests over time in both groups separately, the Friedman repeated measures analysis of variance on ranks followed by Dunn’s method for pairwise multiple comparison were used.

For the binary outcomes (yes/no) (rescue sedation, intra-operative rescue analgesia, noradrenaline administration, postoperative rescue analgesia, atipamezole, Von Frey responders), the relative risk was calculated and the significance verified with chi-square test. The Yates continuity correction was applied.

## 3. Results

A total of 48 rabbits (24 rabbits in group B and 24 rabbits in group M) completed the data collection. One rabbit of the group M had a cardiac arrest after induction of anaesthesia and was successfully resuscitated but excluded from further experimental procedures.

The weight (mean ± SD) of rabbits in group B was 3.30 ± 0.3 kg, and the weight of rabbits in group M was 3.29 ± 0.33 kg. The duration of anaesthesia (mean ± SD) of rabbits in group B was 51.8 ± 9.7 min, and it was 54.46 ± 11.17 min in group M.

For both values, there was no statistically significant difference between treatment groups (two-tailed *p*-value of *t*-test = 0.850 and 0.466, respectively).

### 3.1. Sedation, Intraoperative Nociception, and Respiratory Effects

A total of 15 animals in group B (62.5%) and 18 animals in group M (75%) reached a sedation score ≥ 7 15 min after the SC injection, while 9 (37.5%) and 6 animals (25%), respectively, were deemed to be not adequately sedated. Seven animals in group B (29.2%) and six animals in group M (25%) reached a score ≥ 7 twenty minutes after SC injection, while two animals in group in B (8.3%) and one in group M (4.2%) needed rescue sedation. The likelihood of needing twenty minutes to achieve a satisfactory sedation was 1.5 times higher in group B than in group M, but the difference between groups was not statistically significant (*p* = 0.533).

Median sedation scores were higher in group M [9 (IQR: 7.25–10)] than in group B [8 (IQR 6–9)] (*p* = 0.026) at 15 min after sedation injection and remained significantly higher in group M (n = 6, median 9, IQR 9–10) than in group B (n = 7, median 8.5, IQR 8–9) at 20 min (*p* = 0.005).

Seven animals in group B (29.2%) and five animals in group M (21%) received rescue intraoperative analgesia. The likelihood of needing rescue analgesia in group B was 1.5 times higher than in group M, but the difference was not statistically significant (*p* = 0.739). Intraoperative nociception was detected according to the predefined criteria as presented in Table 1.

Noradrenaline was necessary to correct hypotension in 14 rabbits in group B (58.3%) and in 17 in group M (70.8%). The likelihood of needing noradrenaline was lower (RR = 0.834) in group B than in group M, but the difference was not statistically significant (*p* = 0.546).

The results of the intraoperative blood gas analysis are reported in Table 2.

All rabbits showed respiratory acidosis, the severity of which was not significantly different between groups.

The median PaO_2_/FiO_2_ was 329.5 (IQR = 219.1–420.8) in group B and 390.9 (IQR = 333.895–430.964) in group M. The difference was not statistically significant (*p* = 0.146).

### 3.2. Recovery and Postoperative Pain Assessment

The time (mean ± SD) to achieve sternal recumbency after laryngeal mask removal was 53.7 ± 18 min in group B and 49.75 ± 19 min in group M. There was no statistically significant difference between the treatment groups (two-tailed *p* = 0.466). Atipamezole was administered to 8 animals in group B (33.3%) and 7 animals in group M 29.2%). The likelihood of needing atipamezole in the group B was approaching the likelihood in group M (RR = 0.94), and the difference was not statistically significant (*p* = 1).

Postoperative rescue analgesia was administered to 3 animals in group B (12.5%) and 12 in group M (50%). The likelihood of receiving postoperative analgesia was lower in the group B than in group M (RR = 0.250) and the difference was statistically significant, i.e., the administration of buprenorphine significantly reduced the possibility of receiving rescue postoperative analgesia if compared to methadone (*p* = 0.013). The likelihood of receiving postoperative analgesia was higher in the pool of rabbits that also needed intraoperative analgesia (RR = 1.5), independently of the opioid administered, but the difference was not significant (*p* = 0.476). An RbtGS score < 4 was reached in all rabbits within 2 h after administration of rescue postoperative analgesia (buprenorphine 0.02 mg kg^−1^ SC). The administration of further analgesics was therefore not needed.

RbtGS scores in group B and M are reported in Table 3.

When the Friedman repeated measures analysis of variance on ranks was run, a statistically significant difference of the RbtGS over time was found in group B (*p* = 0.01), but the multiple comparison did not identify any further significance among timepoints in group B. There was no significant difference over time in the group M (*p* = 0.448).

Von Frey filaments thresholds are illustrated in Figure 2. When the Friedman repeated measures analysis of variance on ranks was run, a statistically significant difference of thresholds over time was found in group M (*p* = 0.03), but the multiple comparison did not identify any further significance among timepoints in the group M. There was no difference over time in the group B (*p* = 0.416).

In 16 animals of group B (66.7%) and in 13 animals of group M (54.2%), arterial blood gas was retrieved postoperatively in conditions of comparable FiO_2_ due to the same oxygen supplementation through the same facial mask and estimated as 0.6%. In both group B and M, pH returned to neutrality and PaCO_2_ decreased, with no statistically significant difference between groups. Values are reported in Table 4.

## 4. Discussion

In our trial, buprenorphine and methadone appeared to be two valuable options in the context of balanced anaesthesia for calvaria surgery in New Zealand white rabbits, with buprenorphine showing a more attractive analgesic profile and methadone a better sedative profile. Our results confirmed the hypothesis that methadone, in combination with ketamine and dexmedetomidine, would have provided a deeper sedation than buprenorphine. Clinically, deeper sedation was accompanied by reduced vasoreaction to venous and arterial puncture, facilitating vein and artery cannulation. Interestingly, the rabbits of group M did not require more atipamezole than rabbits of group B, suggesting that the deeper sedation did not affect the speed of recovery.

Our results also confirmed the hypothesis that methadone and buprenorphine would have provided similar respiratory depression. On the other hand, the hypothesis that methadone would have provided more pronounced analgesic effects than buprenorphine was proved wrong, as intraoperatively, a similar number of rabbits in group M and B needed rescue analgesia. Buprenorphine provided longer analgesia than methadone; indeed, animals treated with buprenorphine exhibited a significantly reduced need for additional postoperative analgesia compared to those in the methadone group. Although the use of pure agonists, such as morphine, methadone, hydromorphone, and fentanyl, have been encouraged when moderate to severe pain is present or before an invasive procedure in rabbits [17], very little information is available about the use of µ-pure agonists except for fentanyl [18,19,20], and only one comparative study between buprenorphine and hydromorphone has been carried out in rabbits [21]. In that study, the combination of hydromorphone and alfaxalone induced deeper sedation than the combination of alfaxalone–buprenorphine, while intraoperative nociception was not reported. Our intraoperative findings can be explained by several factors: non equipotent analgesic doses of methadone (a μ-opioid full agonist) and buprenorphine (a µ-opioid partial agonist), potential insufficient absorption of methadone through the injection route, or species-specific pharmacodynamic differences in rabbits at opioids and NMDA receptors. Notably, dosing was based on prior clinical recommendations [12,22] and on authors’ experience, within a broad range of published doses, without proof of equipotency. The choice of SC injection might be also criticised. In female NZWRs, a SC dose of 1 mg kg^−1^ did not achieve plasma concentrationsconsidered analgesic in humans [23]. Although this information should be kept in mind, its interpretation requires caution: first, the minimum analgesic methadone plasma concentration has not been determined in rabbits yet and could be different from other species. Second, the PK of methadone in the context of balanced anaesthesia might differ from the one reported in conscious animals. The availability of buprenorphine was reported to be low after SC injection [24]; therefore, it was suggested that for immediate pain relief, the IM route should be preferred. However, in that trial, the maximal concentration was achieved in females, on average, only 12 min after injection (range of 5–30 min), and in our trial, the surgery on every rabbit started at least 40 min after the SC injection. Undoubtedly, our experience suggested that SC injections were well tolerated in rabbits and resulted in evident sedation and analgesia.

Interestingly, RbtGS values differed between the two groups at 4 and 8 h after achievement of sternal recumbency. Mechanical allodynia was different between the two groups at sternal recumbency and at 6 and 8 h later. At sternal recumbency, the median mechanical allodynia was 26 in group M and 26 in group B. The significant difference between the two groups was likely linked to an outlier. The results of the other timepoints might be explained by the different duration of action of the two opioids. This result is not surprising, as a dose of 2 mg kg^−1^ of methadone injected SC in conscious rabbits provided sedation and analgesia lasting not longer than 150 min [10]. At a dose of 0.01–0.05 mg kg^−1^, the duration of action of buprenorphine is expected to be of 6 to 10 h [22], and its half-life in female conscious rabbits was proven to be 315 ± 139 min (range: 156–495). Pain is a complex, multidimensional experience, and its evaluation cannot be limited to mechanical allodynia. However, we valued the assessment of local sensitivity through the determination of the Von Frey threshold, especially in the early phase of anaesthesia recovery. In this phase, facial pain assessment might be tricky, but the response to Von Frey filaments was always clear and could not be misinterpreted. Electronic Von Frey filaments were deemed useful to assess inflammatory pain in rabbits [25], but little information is available in the literature.

We underestimated the postoperative analgesic effect of buprenorphine, based on the results of a previous observational trial [7] in which buprenorphine was administered SC at 0.02 mg kg^−1^ in conjunction with the same dose of dexmedetomidine and ketamine to male NZWRs for the same surgical purpose. Signs of behavioural postoperative pain were observed in all subjects involved in the trial with a peak of severity at 12 h after recovery from the surgery [7].

The discrepancy in postoperative pain might be explained by the following:(1)The sex of the animals as the response to pain is sex-dependent [26]. Moreover, the PK of SC buprenorphine in male and female NZWRs is markedly different [24].(2)The modality of pain assessment. Males were single-housed before and after the procedure. Single-housed rabbits are generally less active but more restless than are their group-housed counterparts [27] and tend to develop abnormal patterns of locomotion and resting [28]; therefore, falsely high pain scores could have been assigned. Moreover, the trial from Raillard et al. [7] was a descriptive trial and therefore not blinded, and the presence of multiple observers might also have acted as a confounder.(3)The differences in multimodal analgesic management. In our trial, meloxicam was injected IV at a dose of 0.5 mg kg^−1^. Meloxicam is a well used and tested NSAID in companion animals, including rabbits [29]; however, different dosages have been suggested. The most studied way of administration in rabbits is oral, and dosages up to 1.5 mg kg^−1^ have been tested [30]. Dosages of 0.2–0.3 mg Kg^−1^ were initially advised by Carpenter et al. [31], but later it was concluded from the same research group that a dose of 1 mg kg^−1^ PO may be necessary to achieve clinically effective circulating concentrations of meloxicam in rabbits [32]. The bioavailability of meloxicam in rabbits is not known, and the dosage of 1 mg kg^−1^ should be proven safe also if injected IV. Based on the current available data, though, it might be speculated that a dose of 0.3 mg kg^−1^ SC, used also by Raillard et al. [7], is insufficient for preventing postoperative inflammatory pain. Whether buprenorphine and meloxicam have a synergistic effect, as suggested by Goldschlager et al. [33], was not tested in surgical invasive models in rabbits and was not within the focus of our investigation. The role played by local anaesthesia in our results might be significant, but the little information available regarding ropivacaine in rabbits does not allow an extensive discussion or comparison with the effect of lidocaine. A long duration of sensitive nerve block (beyond 300 min) was found after injection of ropivacaine (0.4% up to 0.8 mL kg^−1^) for axillary brachial plexus nerve block in rabbits [34]. Ropivacaine (0.2%) injected intrathecally was deemed effective in 50% of rabbits but safe in terms of hemodynamic variations [35]. It can be expected that the injection of ropivacaine into the periosteum brought about more profound sensitive block than the sole peri-incisional injection of lidocaine. Periosteal single-injection blocks have been used in humans, with success, to reduce distal radius fractures [36,37] and sternal fractures [38], but evidence regarding rabbits is currently lacking. The antinociceptive effect of local analgesia might have contributed to blunt the intraoperative differences between group B and group M. However, the extent of this effect could only be proven by including in the study two further groups receiving methadone or buprenorphine without ropivacaine. Multimodal analgesia in rabbits is still a largely under-researched field [29]; however, it is recommended in many books, articles [4], and conference proceedings on rabbit pain and analgesia. As the beneficial effect of balanced anaesthesia and multimodal analgesia is well known in other species, and there is no evidence against it in rabbits, the authors did decide to stick to this approach; this was also carried out due to ethical concerns.

Although this trial was not designed to evaluate the cardiovascular profile of the two opioids, a similar degree of hypotension was experienced during general anaesthesia in group M and B, and doses of noradrenaline in the range of 0.01–0.03 µg kg^−1^ h^−1^ could restore normotension in all subjects. Drugs such as dexmedetomidine and isoflurane or their combination might have impacted blood pressure, and it is hard to believe that the opioids were the main cause of the hypotension. However, their role cannot be dismissed in this data collection, as blood pressure was not recorded before the induction of general anaesthesia.

In our trial, a similar degree of hypoventilation and respiratory acidosis as the one reported by Raillard et al. [7] was found. As expected, the type of opioid used at the concentration assessed did not have any effect on its severity. Hypoventilation has been reported in rabbits independently of the use of opioids and seems to correlate with the administration of pure oxygen [39]. In our study, the acidosis resolved spontaneously shortly after recovery from anaesthesia. However, the clinical relevance of acidosis is unknown, and the short recovery of normocapnia has been proven in our trial only in ASA 1 rabbits after anaesthesia of short duration Whether this information is valid in other contexts needs to be investigated. Relative hypoxaemia was also recorded in our trial in some subjects. The same issue has been reported with and without the use of opioids [40], and in the absence of the use of alfa_2_ agonists [41], suggesting that it is a collateral effect of sedation in rabbits more than the effect of a specific drug.

This study has some limitations that need to be acknowledged. First, the use of multimodal analgesia could have masked the differences between groups. In particular, the use of dexmedetomidine and ketamine could have prevented detecting intraoperative differences between the two groups. Second, no PK analysis accompanied our clinical observations; therefore, information about bioavailability of both opioids remains empirical. Third, pain faces were not evaluated remotely. The presence of a human observer familiar with the rabbits seemed to reduce scores at the time of greatest pain and increase scores at times of little or no pain, suggesting that remote assessments are preferable if possible [42]. At the time of data collection, a recording system was unfortunately not available, and we tried to minimise the bias through the positioning of a neutral observer out of the rabbits’ field of view.

## 5. Conclusions

In the context of balanced anaesthesia and at the doses tested, the use of methadone resulted in (or produced)better sedative but, overall, lesser analgesic effects, compared to buprenorphine, in female New Zealand white rabbits undergoing calvaria surgery. Particularly, analgesia of longer duration was achieved with the use of buprenorphine. In both treatment groups, individual evaluation of antinociception and postoperative analgesia guaranteed prompt and efficacious correction of their deficits. In the future, it would be worth investigating the analgesic profile of higher doses of methadone and the anti-nociceptive and analgesic profile of IM injections.

## Figures and Tables

**Figure 1 animals-15-01843-f001:**
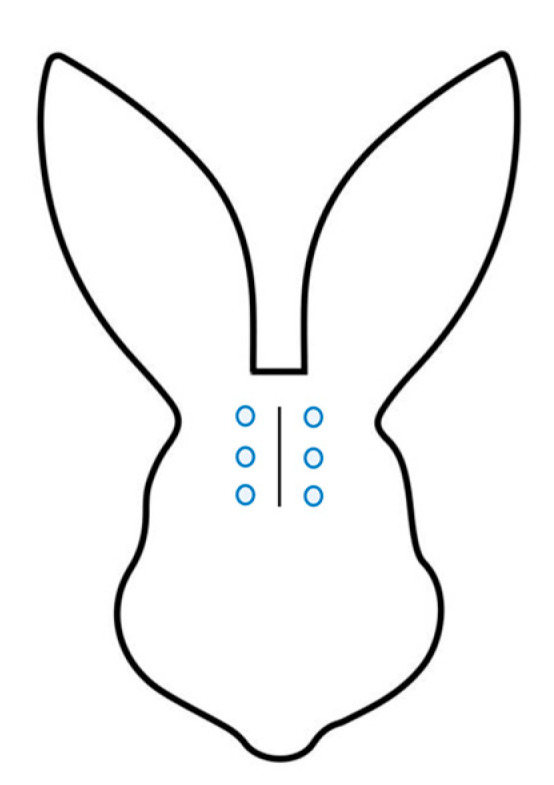
The three landmarks (blue circles) selected along the surgical incision (right and left side) to perform Von Frey tests.

**Figure 2 animals-15-01843-f002:**
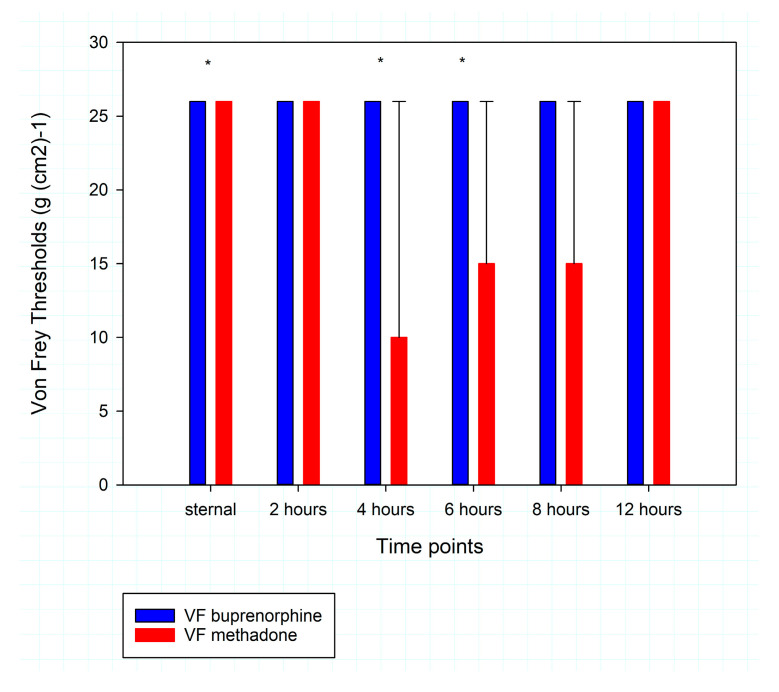
Thresholds of Von Frey filaments testing at the different time points during postoperative recovery. The results are presented as median and IQR. Value of 26 g (cm^2^)^−1^ was assigned to the rabbits that did not react to the thickest filament tested, i.e., 15 g (cm^2^)^−1^. *: statistically significant difference between values recorded in the group that received buprenorphine compared to the group that received methadone.

**Table 1 animals-15-01843-t001:** Indicators of nociception in group M and group B.

Parameter	Number of Animals with 20% Increase from Baseline Group M	Number of Animals with 20% Increase from Baseline Group B
HR	1	1
RR	1	2
BP	0	0
HR/RR	1	2
RR/BP	2	1
HR/BP	0	1

Group M: rabbits receiving methadone; group B: rabbits receiving Buprenorphine; HR: heart rate; RR: respiratory rate; BP: blood pressure.

**Table 2 animals-15-01843-t002:** Intraoperative blood gas values.

Parameter	Group M Median (Min–Max) n = 24	Group B Median (Min–Max) n = 24	*p*-Value
pH	7.289 (7.233–7.379)	7.293 (7.166–7.384)	0.635
FiO_2_	0.95 (86–98)	0.95 (91–100)	0.704
paO_2_ (mmHg)	365.5 (107–433)	313 (90.8–441)	0.132
PaCO_2_ (mmHg)	69.05 (51.5–98.3)	74.1 (57.8–103)	0.122
cBase (Ecf)_c_	7.6 (0.8–15)	9 (3.1–15.6)	0.03 *
cGlucose (mmol/L)	16.65 (12.5–21.7)	17.1 (14.3–20.6)	0.248
cLactate (mmol/L)	0.5 (0.3–1.5)	0.4 (0.2–1.2)	0.453

Group M: rabbits receiving methadone; group B: rabbits receiving buprenorphine; c: concentration; *: statistically significant difference between values recorded in the group that received buprenorphine compared to the group that received methadone.

**Table 3 animals-15-01843-t003:** RbtGS scores once resumed sternal recumbency and at 2 h intervals thereafter.

Time Point	Group M Median (Min–Max)	n = Number of Observations	Group B Median (Min–Max)	n = Number of Observations	*p*-Value (Mann Whitney Runk Sum Test)
sternal	2 (1–3)	24	2 (1–3)	24	0.169
2 h	2 (0–5)	24	2 (0–6)	24	0.228
4 h	2 (1–4)	20	1.5 (0–3)	22	0.023 *
6 h	2 (0–5)	17	1 (0–5)	22	0.108
8 h	2 (1–5)	15	1 (0–3)	21	0.014 *
12 h	1.5 (0–3)	12	1 (0–3)	18	0.471

Group M: rabbits receiving methadone; group B: rabbits receiving buprenorphine; *: statistically significant difference between values recorded in the group that received buprenorphine compared to the group that received methadone.

**Table 4 animals-15-01843-t004:** Postoperative blood gas values.

Parameter	Group M Median (Min–Max) n = 13	Group B Median (Min–Max) n = 16	*p*-Value
pH	7.441 (7.354–7.520)	7.444 (7.388–7.588)	0.854
PaCO_2_ (mmHg)	52.5 (39.9–59.5)	55.7 (45.6–63.2)	0.28
cBase (Ecf)_c_	12.6 (6.75–14.75)	13.85 (12.725–16.325)	0.065
cGlucose (mmol/L)	17.4 (15.95–19.4)	17.95 (17.4–19.8)	0.312
cLactate (mmol/L)	1 (0.55–1.45)	0.7 (0.625–875)	0.376

Group M: rabbits receiving methadone; group B: rabbits receiving buprenorphine; c: concentration.

## Data Availability

Data will be uploaded to the BORIS Portal of the University of Bern once the manuscript is accepted, and these data will be available on request.

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
