# Peer review of "Buprenorphine Versus Methadone in Female New Zealand White Rabbits Undergoing Balanced Anaesthesia for Calvaria Surgery"

_animals, 2025, doi:10.3390/ani15131843_

Round 1
Reviewer 1 Report
Comments and Suggestions for Authors
Summary Comments:
The author addresses an important clinical question regarding the analgesic efficacy of buprenorphine versus methadone in New Zealand White rabbits undergoing calvaria surgery with multimodal anesthetic management. By evaluating both drugs under controlled surgical conditions, the study provides valuable insights into their relative effectiveness for acute pain control in a model relevant to both laboratory and clinical practice.
General Comments:
This study is well-designed and carefully executed, offering a meaningful contribution to the existing knowledge gap regarding opioid efficacy in a controlled surgical and anesthetic setting in New Zealand White rabbits. The use of a standardized surgical model, a multimodal anesthetic protocol, and a systematic pain assessment approach enhances the reliability and translational relevance of the findings.
However, the discussion section would benefit from improved organization and conciseness. The authors are encouraged to more clearly synthesize the key findings and focus their interpretation on potential factors that may explain the finding of methadone did not demonstrate superior analgesic effects compared to buprenorphine during the early postoperative period. While the analgesic effects of some drugs (ropivacaine and meloxicam) are discussed, the potential analgesic contributions of other agents (dexmedetomidine and ketamine) are not discussed. Briefly acknowledging their possible analgesic effects would provide a more comprehensive interpretation of the results for readers. This also raises a question: could a control group receiving a non-opioid-based analgesic protocol have produced similar study outcomes due to both opioids not appropriately reaching the minimal effective analgesic plasma concentration.
Additionally, the limitations could be more effectively presented by consolidating them into a dedicated paragraph. For example, the lack of established minimum effective plasma concentrations for buprenorphine and methadone in this species is a relevant point that warrants clearer emphasis.
Some word choices throughout the manuscript could be revised for greater precision and improved readability.
Overall, this is a valuable and well-conducted study. With minor revisions to the manuscript’s structure and language, its clarity and impact would be further enhanced.
Specific comments:
Thank you for the opportunity to review the manuscript. Overall I have very few comments.
Line 359: Consider replacing ” higher analgesic effect” with” more pronounced analgesic effect".
Line 361-363: Please rephrase the sentence to improve clarity. Consider “Buprenorphine provided a longer duration of analgesic effect than methadone; moreover, animals treated with buprenorphine exhibited a significantly reduced need for additional postoperative analgesic intervention compared to those in the methadone group”
Line 370-377: Kindly reword the sentence to enhance its readability. Consider “Our intraoperative findings may be explained by several factors: non-equipotent dosing between methadone (a µ-opioid full agonist) and buprenorphine (a partial agonist), potential insufficient subcutaneous absorption of methadone, or species-specific pharmacodynamic differences in rabbits. Notably, dosing was based on prior clinical recommendations and investigator experience within the published range, despite the absence of minimal effective plasma concentration.”
Line 377-379: Please rephrase the sentence to make it more readable. “In female NZWR a SC dose of 1 mg kg⁻¹ methadone did not achieve plasma concentrations considered analgesic in humans[22].
Line 380: Please rephrase the term” tangible sedation” is not clear to the readers. Consider “arousable sedation”.
Line 391-395: Please clarify the intention of this statement. In Table 3, the RbtGS scores suggest that all rabbits were comfortable up to 4 hours postoperatively. It's not immediately evident whether the author is emphasizing that most subjects at the sternal time point exhibited lower Von Frey thresholds, aside from a single outlier. Figure 2 (bar chart) does not clearly convey whether there are statistically significant differences at the sternal time point. Please clarify whether the outlier in Group M had a meaningful impact on the results, as this is not addressed in the results section. Consider presenting the data with whiskers blot. Alternatively, including the mean (SD) or median (SD), with and without outliers, for the Von Frey threshold in the results.
Lastly, since this study did not use a methadone dose of 2 mg/kg, it would not be appropriate to compare the effective duration of methadone with studies using different dosing regimens
Line 405-424: Please rephrase the paragraph for clarity and conciseness. The original text is somewhat repetitive, especially in its comparison between studies. Additionally, the paragraph is difficult to follow, and it is unclear whether the author aims to highlight a disagreement regarding the efficacy of buprenorphine in similar research settings or to address a limitation in pain assessment within the different study.
Line 448-449: Please consider replacing “blunting effect of local analgesia” with” modulatory effect of local anesthesia on nociceptive responses.”
Line 449-456: The authors may consider briefly discussing how the use of multimodal anaesthesia—including dexmedetomidine, ketamine, locoregional anaesthesia, and meloxicam in the intraoperative and postoperative analgesia protocol—could have influenced the assessment of intraoperative and postoperative pain. This multimodal approach may have potentially masked any observable differences in analgesic efficacy between the methadone and buprenorphine groups. Addressing this point would enhance the interpretation of the findings and clarify the limitations related to detecting subtle differences in opioid efficacy.
Line 460-462: The statement regarding the potential role of opioids in intraoperative hypotension lacks clarity for the reader. Specifically, it is unclear why the discussion focuses on opioids as a possible cause of hypotension, while the cardiovascular effects of other agents used—such as the alpha-2 agonist dexmedetomidine—are not mentioned, despite their well-documented hemodynamic impacts. While the cardiovascular effects of these agents may fall outside the primary scope of the study, briefly acknowledging their potential contributions would offer a more comprehensive interpretation of the intraoperative hemodynamic findings.
Line 472-473: I recommend replacing “evaluation of pain faces” with the term “pain assessment.” Additionally, it may be beneficial to consolidate all study limitations into a single section. For example, current literature does not clearly establish the minimal effective plasma concentration of methadone in rabbits, and the use of a multimodal anesthesia protocol may increase the risk of Type II errors when comparing intraoperative/postoperative outcomes between groups.
Line 483: Please correct the typo of treatment.
The manuscript is overall well-constructed with strong scientific writing. However, the discussion section would benefit from minor language improvements to enhance clarity and readability.
Author Response
The author addresses an important clinical question regarding the analgesic efficacy of buprenorphine versus methadone in New Zealand White rabbits undergoing calvaria surgery with multimodal anesthetic management. By evaluating both drugs under controlled surgical conditions, the study provides valuable insights into their relative effectiveness for acute pain control in a model relevant to both laboratory and clinical practice.
General Comments:
This study is well-designed and carefully executed, offering a meaningful contribution to the existing knowledge gap regarding opioid efficacy in a controlled surgical and anesthetic setting in New Zealand White rabbits. The use of a standardized surgical model, a multimodal anesthetic protocol, and a systematic pain assessment approach enhances the reliability and translational relevance of the findings.
Thank you for your appreciation and the time you dedicated to this review. We appreciated your valuable inputs.
However, the discussion section would benefit from improved organization and conciseness. The authors are encouraged to more clearly synthesize the key findings and focus their interpretation on potential factors that may explain the finding of methadone did not demonstrate superior analgesic effects compared to buprenorphine during the early postoperative period. While the analgesic effects of some drugs (ropivacaine and meloxicam) are discussed, the potential analgesic contributions of other agents (dexmedetomidine and ketamine) are not discussed. Briefly acknowledging their possible analgesic effects would provide a more comprehensive interpretation of the results for readers. This also raises a question: could a control group receiving a non-opioid-based analgesic protocol have produced similar study outcomes due to both opioids not appropriately reaching the minimal effective analgesic plasma concentration.
We tried to make the discussion more concise according to your suggestion. We understand your point. As note, ropivacaine and meloxicam were discussed in relation to the differences with the previous trial of Raillard et al. We appreciate anyway the need to cite the role of multimodal analgesia as potential masking factor and mention ketamine and dexmedetomidine (paragraph of limitation).
Additionally, the limitations could be more effectively presented by consolidating them into a dedicated paragraph. For example, the lack of established minimum effective plasma concentrations for buprenorphine and methadone in this species is a relevant point that warrants clearer emphasis.
Some word choices throughout the manuscript could be revised for greater precision and improved readability.
Overall, this is a valuable and well-conducted study. With minor revisions to the manuscript’s structure and language, its clarity and impact would be further enhanced.
Specific comments:
Thank you for the opportunity to review the manuscript. Overall I have very few comments.
Line 359: Consider replacing ” higher analgesic effect” with” more pronounced analgesic effect".
Word replaced as suggested.
Line 361-363: Please rephrase the sentence to improve clarity. Consider “Buprenorphine provided a longer duration of analgesic effect than methadone; moreover, animals treated with buprenorphine exhibited a significantly reduced need for additional postoperative analgesic intervention compared to those in the methadone group”
Sentence rephrased based on your suggestion.
Line 370-377: Kindly reword the sentence to enhance its readability. Consider “Our intraoperative findings may be explained by several factors: non-equipotent dosing between methadone (a µ-opioid full agonist) and buprenorphine (a partial agonist), potential insufficient subcutaneous absorption of methadone, or species-specific pharmacodynamic differences in rabbits. Notably, dosing was based on prior clinical recommendations and investigator experience within the published range, despite the absence of minimal effective plasma concentration.”
The sentence has been rephrased according to your suggestions. However, our specific point in the last part of the sentence was not that we do not know the minimal effective plasma concentration of this drug in rabbits, but we do not know whether 0.02 mg/kg buprenorphine SC is equipotent to 0.3 mg/kg SC of methadone. Therefore, the last part of the sentence was not modified.
Line 377-379: Please rephrase the sentence to make it more readable. “In female NZWR a SC dose of 1 mg kg⁻¹ methadone did not achieve plasma concentrations considered analgesic in humans[22].
Sentence rephrased based on your suggestion.
Line 380: Please rephrase the term” tangible sedation” is not clear to the readers. Consider “arousable sedation”.
The word tangible was substituted with evident, as I fear “arousable” was not what we meant.
Line 391-395: Please clarify the intention of this statement. In Table 3, the RbtGS scores suggest that all rabbits were comfortable up to 4 hours postoperatively. It's not immediately evident whether the author is emphasizing that most subjects at the sternal time point exhibited lower Von Frey thresholds, aside from a single outlier. Figure 2 (bar chart) does not clearly convey whether there are statistically significant differences at the sternal time point. Please clarify whether the outlier in Group M had a meaningful impact on the results, as this is not addressed in the results section. Consider presenting the data with whiskers blot. Alternatively, including the mean (SD) or median (SD), with and without outliers, for the Von Frey threshold in the results.
Thank you for raising this point. We rephrased this sentence as we realised it could have been misleading for the reader. We also modified the legend of figure 2. The plots are representing median and IQR. The median of Von Frey filament value at sternal was the same in both groups (and in both groups did not indicate mechanical allodynia). This is now clearly stated in the text. Only in one animal values suggesting allodynia were collected.
Lastly, since this study did not use a methadone dose of 2 mg/kg, it would not be appropriate to compare the effective duration of methadone with studies using different dosing regimens
We rephrased this sentence. We agree that a comparison would not be appropriate. We meant that if a higher dose is providing analgesia for less than 3 hours, it is not surprising the duration we found in our study.
Line 405-424: Please rephrase the paragraph for clarity and conciseness. The original text is somewhat repetitive, especially in its comparison between studies. Additionally, the paragraph is difficult to follow, and it is unclear whether the author aims to highlight a disagreement regarding the efficacy of buprenorphine in similar research settings or to address a limitation in pain assessment within the different study.
We trimmed the paragraph and underlined the notable differences between the two studies, which could have caused the different results achieved. Thank you for raising this point.
Line 448-449: Please consider replacing “blunting effect of local analgesia” with” modulatory effect of local anesthesia on nociceptive responses.”
The sentence has been rephrased.
Line 449-456: The authors may consider briefly discussing how the use of multimodal anaesthesia—including dexmedetomidine, ketamine, locoregional anaesthesia, and meloxicam in the intraoperative and postoperative analgesia protocol—could have influenced the assessment of intraoperative and postoperative pain. This multimodal approach may have potentially masked any observable differences in analgesic efficacy between the methadone and buprenorphine groups. Addressing this point would enhance the interpretation of the findings and clarify the limitations related to detecting subtle differences in opioid efficacy.
This was discussed in lines 472-480. We pointed it out also in the paragraph of limitations.
Line 460-462: The statement regarding the potential role of opioids in intraoperative hypotension lacks clarity for the reader. Specifically, it is unclear why the discussion focuses on opioids as a possible cause of hypotension, while the cardiovascular effects of other agents used—such as the alpha-2 agonist dexmedetomidine—are not mentioned, despite their well-documented hemodynamic impacts. While the cardiovascular effects of these agents may fall outside the primary scope of the study, briefly acknowledging their potential contributions would offer a more comprehensive interpretation of the intraoperative hemodynamic findings.
We understand your concern and we agree on your statement. The clinical experience suggested that hypotension is most likely due to the combination of alfa-2 agonists and isoflurane. However, this is not a result of this study, therefore we simply cited the potential role of these drugs in the correct version.
Line 472-473: I recommend replacing “evaluation of pain faces” with the term “pain assessment.” Additionally, it may be beneficial to consolidate all study limitations into a single section. For example, current literature does not clearly establish the minimal effective plasma concentration of methadone in rabbits, and the use of a multimodal anesthesia protocol may increase the risk of Type II errors when comparing intraoperative/postoperative outcomes between groups.
We consolidated the limitations in a final paragraph as suggested. We would prefer to keep the definition of “pain faces” evaluation as the limitation is referred to the Grimace Scale. Quantitative pain evaluation requires the presence of an operator in the vicinity, and more than a limitation of the study is a con of the technique per se.
Line 483: Please correct the typo of treatment.
Corrected. Apologies.
Reviewer 2 Report
Comments and Suggestions for Authors
Dear Auhtors,
Congratulations on the elaborate paper. The text is very clear, pleasant and flowing. I have carefully reread the paper several times and notice a lot of attention to all the sections. The structuring of the paper is well executed, material and methods are clear and comprehensive. I would only point out that on line 155 there is an inscription in brackets that probably needs to be deleted. Immediately after a sentence point there is a double space which I would ask you to correct. The discussions in my opinion are a bit too long and this could make the reader lose interest. Otherwise, I renew my ocmpliemnts, nice work.
Author Response
Dear reviewer,
thank you very much for the time and the attention you dedicated to our manuscript. We are pleased that you found the study well conducted. We corrected the tape mistakes you pointed out. We tried to improve the conciseness of the discussion and avoid some redundancies, as suggested also by another reviewer. Since we were asked to point out some other little aspects, I am not sure whether the discussion is really shorter than before, but I hope the readability is improved. Once more, thank you for your work.
Reviewer 3 Report
Comments and Suggestions for Authors
Introduction
Refer to the regulations governing animal experimentation in your country.
2.1. Ethical Approval
Indicate in the paper the state regulations that apply to animal experimentation.
2.2 Animals
Indicate the geographic coordinates of the study site in the paper.
Indicate the age and origin of the animals in the paper.
Indicate the number of animals per group in the paper. Indicate the cage dimensions in the paper.
Justify in the paper why only females were used in the experiment.
Were antibiotics used? State this in the paper.
2.3. Anesthesia Management
Indicate in the paper whether the anesthesiologist was authorized to perform animal experimentation.
How were the animals handled? It is important to mention in the literature how the animals were handled to avoid injury to both the animals and the people handling them. Were any of the recommendations followed, for example, by Chaptel et al. 2015 (Chapel, J.; Benedito, J.; Hernández, J.; Pereira, V.; Domínguez, R.; Castillo, C. Handling and restraint techniques for domestic rabbits. Consult. Difusión Vet. 2015, 221, 47–54.)?
Author Response
Introduction
Refer to the regulations governing animal experimentation in your country.
We included this information in the text
2.1. Ethical Approval
Indicate in the paper the state regulations that apply to animal experimentation.
It is now indicated after the ethical permission number.
2.2 Animals
Indicate the geographic coordinates of the study site in the paper.
The study was carried out at the University of Bern, as specified. The facilities are indicated but further details are not provided for the safety of the personnel working in the Facilities.
Indicate the age and origin of the animals in the paper.
Added in the text
Indicate the number of animals per group in the paper. Indicate the cage dimensions in the paper.
Information added in the text
Justify in the paper why only females were used in the experiment.
The decision of including only females was unfortunately not in the Authors’ hands. A previous data collection in males was performed and the researchers wanted to continue their experiment in females. This reason has been reported in the text.
Were antibiotics used? State this in the paper.
Antibiotics were used as already indicated in the text (lines 145-146)
2.3. Anesthesia Management
Indicate in the paper whether the anesthesiologist was authorized to perform animal experimentation.
The veterinary anaesthesiologist is the Head of the Experimental Surgery Facility, and his name was included in the list of personnel involved in the experiment. This information is available in the ethical permission.
How were the animals handled? It is important to mention in the literature how the animals were handled to avoid injury to both the animals and the people handling them. Were any of the recommendations followed, for example, by Chaptel et al. 2015 (Chapel, J.; Benedito, J.; Hernández, J.; Pereira, V.; Domínguez, R.; Castillo, C. Handling and restraint techniques for domestic rabbits. Consult. Difusión Vet. 2015, 221, 47–54.)?
I am not sure that it was clear that the anaesthetist was a veterinarian by training. The rabbits were handled in the Central Animal Facility according to the principles of humane handling and the recommendations of Bradbury AG, Dickens GJ. Appropriate handling of pet rabbits: a literature review. J Small Anim Pract. 2016 Oct;57(10):503-509 were followed. Information has been added to the text.
Reviewer 4 Report
Comments and Suggestions for Authors
Dear Authors,
Congratulations on the writing of this very clear manuscript. To further improve the text, I would like to offer a few minor comments, as listed below.
Line 53: “unknowly painful” → with an unpredictable degree of pain
Line 64: “dose- response curve” → dose-response curve
Line 121: A period is missing at the end of the sentence.
Line 140: Please indicate the dose of ropivacaine administered in the text.
Line 155: The brand and model of the monitor are missing.
Line 163: Please specify the route of administration for ketamine used as rescue analgesia.
Line 170: What was the rationale for choosing the specific dose of 0.4 mg/kg atipamezole, after dexmedetomidine at 0.1 mg/kg?
Line 179: “were filled” → was filled
Line 193: For what type of analysis?
Line 235: Which scoring sheet was used?
Lines 277–279: To provide a more immediate understanding of the sedation scores, I suggest adding the percentage for each group. For example: fifteen animals in group B (62.5%). Consider doing the same throughout the Results section.
Lines 295–298: The number of cases in which norepinephrine administration was required appears to be quite high. Considering that this drug is unfortunately not often implemented in many clinical settings—either due to a lack of appropriate equipment or experience with the species—this observation is particularly relevant in the overall evaluation of the anesthetic protocol, which may be applicable in a clinical context. For this reason, I believe it would be highly beneficial for the reader if some physiological parameters (HR, RR, SAP, DAP, MAP) were reported in a table, even if only as median and range, to provide a broader understanding of the effects of the protocol.
Line 378: Does this dose refer to methadone?
Author Response
Dear Authors,
Congratulations on the writing of this very clear manuscript. To further improve the text, I would like to offer a few minor comments, as listed below.
Thank you for the time you dedicated to review our manuscript and for the valuable suggestions you made.
Line 53: “unknowly painful” → with an unpredictable degree of pain
Sentence modified as suggested
Line 64: “dose- response curve” → dose-response curve
Apologies for the oversight
Line 121: A period is missing at the end of the sentence.
Apologies for the oversight
Line 140: Please indicate the dose of ropivacaine administered in the text.
Information added in the text
Line 155: The brand and model of the monitor are missing.
Apologies, it was meant to be there and got forgotten.
Line 163: Please specify the route of administration for ketamine used as rescue analgesia.
Information added in the text
Line 170: What was the rationale for choosing the specific dose of 0.4 mg/kg atipamezole, after dexmedetomidine at 0.1 mg/kg?
We wanted to administer a pre-determined dose of atipamezole and for animals between 3 and 4 Kg the advised dose of atipamezole to antagonize IM atipamezole is 350-400 mcg Kg-1.
Line 179: “were filled” → was filled
The subject is 300 microliters, we think the plural is correct.
Line 193: For what type of analysis?
Blood gas analysis, the information was added in the text
Line 235: Which scoring sheet was used?
A behavioural score sheet including categories as; appetite, defecation, demeanour, movement, appearance, weight, social interaction. As data after the 24 hours are not presented in this manuscript, detailed information about this score sheet do not seem relevant to the authors.
Lines 277–279: To provide a more immediate understanding of the sedation scores, I suggest adding the percentage for each group. For example: fifteen animals in group B (62.5%). Consider doing the same throughout the Results section.
Information added in the text.
Lines 295–298: The number of cases in which norepinephrine administration was required appears to be quite high. Considering that this drug is unfortunately not often implemented in many clinical settings—either due to a lack of appropriate equipment or experience with the species—this observation is particularly relevant in the overall evaluation of the anesthetic protocol, which may be applicable in a clinical context. For this reason, I believe it would be highly beneficial for the reader if some physiological parameters (HR, RR, SAP, DAP, MAP) were reported in a table, even if only as median and range, to provide a broader understanding of the effects of the protocol.
The cardiovascular characterization of the protocols falls out from the aims of this manuscript and such a table might be in our opinion misleading. As a clinical information, the blood pressure decreased when isoflurane was started. As we had a cut-off of 60 mmHg MAP to start inotropes/vasopressors, and we measured IBP, we intervened already with mild hypotension (MAP = 55 mmHg) and quickly. As result, measured blood pressures are similar throughout the procedure, because of our pharmacological manipulation. HR tended also to decrease after administration of norepinephrine (the same happens with dobutamine) and either the phenomenon is described fully or might be misleading. The message we wanted to pass the reader was that it is frequent to experience hypotension during general inhalational anaesthesia. As mentioned in the discussion, it is hard to believe that this depends on opioids. Clinically, one sees clearly that the hypotension starts with administration of isoflurane. If you think it might be beneficial to the reader, RR could be reported. However, tidal volume was not measured, so the information might not be of great help.
Line 378: Does this dose refer to methadone?
This part of the discussion has been modified following the suggestions of another reviewer. I hope you can find the answer to your question in the implemented text.